# ENSEMBLE ADVERSARIAL TRAINING: ATTACKS AND DEFENSES

**Florian Tramèr**
Stanford University
tramer@cs.stanford.edu

**Alexey Kurakin**
Google Brain
kurakin@google.com

**Nicolas Papernot**[*]
Pennsylvania State University
ngp5056@cse.psu.edu

**Ian Goodfellow**
Google Brain
goodfellow@google.com

**Dan Boneh**
Stanford University
dabo@cs.stanford.edu

**Patrick McDaniel**
Pennsylvania State University
mcdaniel@cse.psu.edu

## ABSTRACT

Adversarial examples are perturbed inputs designed to fool machine learning models. Adversarial training injects such examples into training data to increase robustness. To scale this technique to large datasets, perturbations are crafted using fast *single-step* methods that maximize a linear approximation of the model's loss.

We show that this form of adversarial training converges to a degenerate global minimum, wherein small curvature artifacts near the data points obfuscate a linear approximation of the loss. The model thus learns to generate weak perturbations, rather than defend against strong ones. As a result, we find that adversarial training remains vulnerable to black-box attacks, where we *transfer* perturbations computed on undefended models, as well as to a powerful novel single-step attack that escapes the non-smooth vicinity of the input data via a small random step.

We further introduce *Ensemble Adversarial Training*, a technique that augments training data with perturbations transferred from other models. On ImageNet, Ensemble Adversarial Training yields models with strong robustness to black-box attacks. In particular, our most robust model won the first round of the *NIPS 2017 competition on Defenses against Adversarial Attacks* (Kurakin et al., 2017c).

## 1 INTRODUCTION

Machine learning (ML) models are often vulnerable to *adversarial examples*, maliciously perturbed inputs designed to mislead a model at test time (Biggio et al., 2013; Szegedy et al., 2013; Goodfellow et al., 2014b; Papernot et al., 2016a). Furthermore, Szegedy et al. (2013) showed that these inputs *transfer* across models: the same adversarial example is often misclassified by different models, thus enabling simple *black-box* attacks on deployed models (Papernot et al., 2017; Liu et al., 2017).

*Adversarial training* (Szegedy et al., 2013) increases robustness by augmenting training data with adversarial examples. Madry et al. (2017) showed that adversarially trained models can be made robust to *white-box* attacks (i.e., with knowledge of the model parameters) if the perturbations computed during training closely maximize the model's loss. However, prior attempts at scaling this approach to ImageNet-scale tasks (Deng et al., 2009) have proven unsuccessful (Kurakin et al., 2017b).

It is thus natural to ask whether it is possible, at scale, to achieve robustness against the class of *black-box* adversaries Towards this goal, Kurakin et al. (2017b) adversarially trained an Inception v3 model (Szegedy et al., 2016b) on ImageNet using a "single-step" attack based on a linearization of the model's loss (Goodfellow et al., 2014b). Their trained model is robust to single-step perturbations but remains vulnerable to more costly "multi-step" attacks. Yet, Kurakin et al. (2017b) found that these attacks fail to reliably transfer between models, and thus concluded that the robustness of their model should extend to black-box adversaries. Surprisingly, we show that this is not the case.

---

[*]Part of the work was done while the author was at Google Brain.

We demonstrate, formally and empirically, that *adversarial training with single-step methods admits a degenerate global minimum*, wherein the model's loss can not be reliably approximated by a linear function. Specifically, we find that the model's decision surface exhibits sharp curvature near the data points, thus degrading attacks based on a single gradient computation. In addition to the model of Kurakin et al. (2017b), we reveal similar overfitting in an adversarially trained Inception ResNet v2 model (Szegedy et al., 2016a), and a variety of models trained on MNIST (LeCun et al., 1998).

We harness this result in two ways. First, we show that adversarially trained models using single-step methods remain vulnerable to simple attacks. For black-box adversaries, we find that perturbations crafted on an undefended model often transfer to an adversarially trained one. We also introduce a simple yet powerful single-step attack that applies a small *random* perturbation—to escape the non-smooth vicinity of the data point—before linearizing the model's loss. While seemingly weaker than the Fast Gradient Sign Method of Goodfellow et al. (2014b), our attack significantly outperforms it for a same perturbation norm, for models trained *with or without adversarial training*.

Second, we propose *Ensemble Adversarial Training*, a training methodology that incorporates perturbed inputs transferred from other *pre-trained models*. Our approach *decouples* adversarial example generation from the parameters of the trained model, and increases the *diversity* of perturbations seen during training. We train Inception v3 and Inception ResNet v2 models on ImageNet that exhibit increased robustness to adversarial examples transferred from other holdout models, using various single-step and multi-step attacks (Goodfellow et al., 2014b; Carlini & Wagner, 2017a; Kurakin et al., 2017a; Madry et al., 2017). We also show that our methods globally reduce the *dimensionality* of the space of adversarial examples (Tramèr et al., 2017). Our Inception ResNet v2 model won the first round of the NIPS 2017 competition on Defenses Against Adversarial Attacks (Kurakin et al., 2017c), where it was evaluated on other competitors' attacks in a black-box setting.[1]

## 2 RELATED WORK

Various defensive techniques against adversarial examples in deep neural networks have been proposed (Gu & Rigazio, 2014; Luo et al., 2015; Papernot et al., 2016c; Nayebi & Ganguli, 2017; Cisse et al., 2017) and many remain vulnerable to adaptive attackers (Carlini & Wagner, 2017a;b; Baluja & Fischer, 2017). Adversarial training (Szegedy et al., 2013; Goodfellow et al., 2014b; Kurakin et al., 2017b; Madry et al., 2017) appears to hold the greatest promise for learning robust models.

Madry et al. (2017) show that adversarial training on MNIST yields models that are robust to white-box attacks, if the adversarial examples used in training closely maximize the model's loss. Moreover, recent works by Sinha et al. (2018), Raghunathan et al. (2018) and Kolter & Wong (2017) even succeed in providing *certifiable* robustness for small perturbations on MNIST. As we argue in Appendix C, the MNIST dataset is peculiar in that there exists a simple "closed-form" *denoising* procedure (namely feature binarization) which leads to similarly robust models *without adversarial training*. This may explain why robustness to white-box attacks is hard to scale to tasks such as ImageNet (Kurakin et al., 2017b). We believe that the existence of a simple robust baseline for MNIST can be useful for understanding some *limitations* of adversarial training techniques.

Szegedy et al. (2013) found that adversarial examples transfer between models, thus enabling black-box attacks on deployed models. Papernot et al. (2017) showed that black-box attacks could succeed with no access to training data, by exploiting the target model's predictions to *extract* (Tramèr et al., 2016) a surrogate model. Some prior works have hinted that adversarially trained models may remain vulnerable to black-box attacks: Goodfellow et al. (2014b) found that an adversarial maxout network on MNIST has slightly higher error on transferred examples than on white-box examples. Papernot et al. (2017) further showed that a model trained on small perturbations can be evaded by transferring perturbations of larger magnitude. Our finding that adversarial training degrades the accuracy of linear approximations of the model's loss is as an instance of a *gradient-masking* phenomenon (Papernot et al., 2016b), which affects other defensive techniques (Papernot et al., 2016c; Carlini & Wagner, 2017a; Nayebi & Ganguli, 2017; Brendel & Bethge, 2017; Athalye et al., 2018).

---

[1] We publicly released our model after the first round, and it could thereafter be targeted using white-box attacks. Nevertheless, a majority of the top submissions in the final round, e.g. (Xie et al., 2018) built upon our released model.

# 3 THE ADVERSARIAL TRAINING FRAMEWORK

We consider a classification task with data $x \in [0, 1]^d$ and labels $y_{\text{true}} \in \mathbb{Z}_k$ sampled from a distribution $\mathcal{D}$. We identify a model with an hypothesis $h$ from a space $\mathcal{H}$. On input $x$, the model outputs class scores $h(x) \in \mathbb{R}^k$. The loss function used to train the model, e.g., cross-entropy, is $L(h(x), y)$.

## 3.1 THREAT MODEL

For some target model $h \in \mathcal{H}$ and inputs $(x, y_{\text{true}})$ the adversary's goal is to find an *adversarial example* $x^{\text{adv}}$ such that $x^{\text{adv}}$ and $x$ are "close" yet the model misclassifies $x^{\text{adv}}$. We consider the well-studied class of $\ell_\infty$ bounded adversaries (Goodfellow et al., 2014b; Madry et al., 2017) that, given some *budget* $\epsilon$, output examples $x^{\text{adv}}$ where $\|x^{\text{adv}} - x\|_\infty \leq \epsilon$. As we comment in Appendix C.1, $\ell_\infty$ robustness is of course not an end-goal for secure ML. We use this standard model to showcase limitations of prior adversarial training methods, and evaluate our proposed improvements.

We distinguish between *white-box* adversaries that have access to the target model's parameters (i.e., $h$), and *black-box* adversaries with only partial information about the model's inner workings. Formal definitions for these adversaries are in Appendix A. Although security against white-box attacks is the stronger notion (and the one we ideally want ML models to achieve), black-box security is a reasonable and more tractable goal for deployed ML models.

## 3.2 ADVERSARIAL TRAINING

Following Madry et al. (2017), we consider an adversarial variant of standard Empirical Risk Minimization (ERM), where our aim is to minimize the risk over adversarial examples:

$$h^* = \underset{h \in \mathcal{H}}{\arg\min} \; \underset{(x, y_{\text{true}}) \sim \mathcal{D}}{\mathbb{E}} \left[ \max_{\|x^{\text{adv}} - x\|_\infty \leq \epsilon} L(h(x^{\text{adv}}), y_{\text{true}}) \right] . \tag{1}$$

Madry et al. (2017) argue that adversarial training has a natural interpretation in this context, where a given attack (see below) is used to approximate solutions to the inner maximization problem, and the outer minimization problem corresponds to training over these examples. Note that the original formulation of adversarial training (Szegedy et al., 2013; Goodfellow et al., 2014b), which we use in our experiments, trains on both the "clean" examples $x$ and adversarial examples $x^{\text{adv}}$.

We consider three algorithms to generate adversarial examples with bounded $\ell_\infty$ norm. The first two are *single-step* (i.e., they require a single gradient computation); the third is *iterative*—it computes multiple gradient updates. We enforce $x^{\text{adv}} \in [0, 1]^d$ by clipping all components of $x^{\text{adv}}$.

**Fast Gradient Sign Method (FGSM).** This method (Goodfellow et al., 2014b) linearizes the inner maximization problem in (1):

$$x^{\text{adv}}_{\text{FGSM}} := x + \varepsilon \cdot \texttt{sign} \left( \nabla_x L(h(x), y_{\text{true}}) \right) . \tag{2}$$

**Single-Step Least-Likely Class Method (Step-LL).** This variant of FGSM introduced by Kurakin et al. (2017a;b) targets the least-likely class, $y_{\text{LL}} = \arg\min\{h(x)\}$:

$$x^{\text{adv}}_{\text{LL}} := x - \varepsilon \cdot \texttt{sign} \left( \nabla_x L(h(x), y_{\text{LL}}) \right) . \tag{3}$$

Although this attack only indirectly tackles the inner maximization in (1), Kurakin et al. (2017b) find it to be the most effective for adversarial training on ImageNet.

**Iterative Attack (I-FGSM or Iter-LL).** This method iteratively applies the FGSM or Step-LL $k$ times with step-size $\alpha \geq \epsilon/k$ and projects each step onto the $\ell_\infty$ ball of norm $\epsilon$ around $x$. It uses projected gradient descent to solve the maximization in (1). For fixed $\epsilon$, iterative attacks induce higher error rates than single-step attacks, but *transfer* at lower rates (Kurakin et al., 2017a;b).

## 3.3 A DEGENERATE GLOBAL MINIMUM FOR SINGLE-STEP ADVERSARIAL TRAINING

When performing adversarial training with a single-step attack (e.g., the FGSM or Step-LL methods above), we approximate Equation (1) by replacing the solution to the inner maximization problem

in with the output of the single-step attack (e.g., $x_{\text{FGSM}}^{\text{adv}}$ in (2)). That is, we solve

$$h^* = \arg\min_{h \in \mathcal{H}} \quad \mathbb{E}_{(x,y_{\text{true}}) \sim \mathcal{D}} \left[ L(h(x_{\text{FGSM}}^{\text{adv}}), y_{\text{true}}) \right] . \tag{4}$$

For model families $\mathcal{H}$ with high expressive power, this alternative optimization problem admits at least two substantially different global minima $h^*$:

- For an input $x$ from $\mathcal{D}$, there is no $x^{\text{adv}}$ close to $x$ (in $\ell_\infty$ norm) that induces a high loss. That is,

$$L(h^*(x_{\text{FGSM}}^{\text{adv}}), y_{\text{true}}) \approx \max_{\|x^{\text{adv}} - x\|_\infty \leq \epsilon} L(h^*(x^{\text{adv}}), y_{\text{true})}] \approx 0 . \tag{5}$$

  In other words, $h^*$ is robust to all $\ell_\infty$ bounded perturbations.
- The minimizer $h^*$ is a model for which the approximation method underlying the attack (i.e., linearization in our case) poorly fits the model's loss function. That is,

$$L(h^*(x_{\text{FGSM}}^{\text{adv}}), y_{\text{true}}) \ll \max_{\|x^{\text{adv}} - x\|_\infty \leq \epsilon} L(h^*(x^{\text{adv}}), y_{\text{true}})] . \tag{6}$$

  Thus the attack when applied to $h^*$ produces samples $x^{\text{adv}}$ that are far from optimal.

Note that this second "degenerate" minimum can be more subtle than a simple case of overfitting to samples produced from single-step attacks. Indeed, we show in Section 4.1 that single-step attacks applied to adversarially trained models create "adversarial" examples *that are easy to classify even for undefended models*. Thus, adversarial training does not simply learn to resist the particular attack used during training, but actually to make that attack perform worse overall. This phenomenon relates to the notion of *Reward Hacking* (Amodei et al., 2016) wherein an agent maximizes its formal objective function via unintended behavior that fails to captures the designer's true intent.

### 3.4 ENSEMBLE ADVERSARIAL TRAINING

The degenerate minimum described in Section 3.3 is attainable because the learned model's parameters influence the quality of both the minimization and maximization in (1). One solution is to use a stronger adversarial example generation process, at a high performance cost (Madry et al., 2017). Alternatively, Baluja & Fischer (2017) suggest training an adversarial *generator* model as in the GAN framework (Goodfellow et al., 2014a). The power of this generator is likely to require careful tuning, to avoid similar degenerate minima (where the generator or classifier overpowers the other).

We propose a conceptually simpler approach to *decouple* the generation of adversarial examples from the model being trained, while simultaneously drawing an explicit connection with robustness to black-box adversaries. Our method, which we call *Ensemble Adversarial Training*, augments a model's training data with adversarial examples crafted on other *static pre-trained* models. Intuitively, as adversarial examples transfer between models, perturbations crafted on an external model are good approximations for the maximization problem in (1). Moreover, the learned model can not influence the "strength" of these adversarial examples. As a result, minimizing the training loss implies increased robustness to black-box attacks from some set of models.

**Domain Adaptation with multiple sources.** We can draw a connection between Ensemble Adversarial Training and *multiple-source Domain Adaptation* (Mansour et al., 2009; Zhang et al., 2012). In Domain Adaptation, a model trained on data sampled from one or more *source distributions* $\mathcal{S}_1, \ldots, \mathcal{S}_k$ is evaluated on samples $x$ from a different *target distribution* $\mathcal{T}$.

Let $\mathcal{A}_i$ be an adversarial distribution obtained by sampling $(x, y_{\text{true}})$ from $\mathcal{D}$, computing an adversarial example $x^{\text{adv}}$ for some model such that $\|x^{\text{adv}} - x\|_\infty \leq \epsilon$, and outputting $(x^{\text{adv}}, y_{\text{true}})$. In Ensemble Adversarial Training, the source distributions are $\mathcal{D}$ (the clean data) and $\mathcal{A}_1, \ldots, \mathcal{A}_k$ (the attacks overs the currently trained model and the static pre-trained models). The target distribution takes the form of an unseen black-box adversary $\mathcal{A}^*$. Standard generalization bounds for Domain Adaptation (Mansour et al., 2009; Zhang et al., 2012) yield the following result.

**Theorem 1** (informal). *Let $h^* \in \mathcal{H}$ be a model learned with Ensemble Adversarial Training and static black-box adversaries $\mathcal{A}_1, \ldots, \mathcal{A}_k$. Then, if $h^*$ is robust against the black-box adversaries $\mathcal{A}_1, \ldots \mathcal{A}_k$ used at training time, then $h^*$ has bounded error on attacks from a future black-box adversary $\mathcal{A}^*$, if $\mathcal{A}^*$ is not "much stronger", on average, than the static adversaries $\mathcal{A}_1, \ldots, \mathcal{A}_k$.*

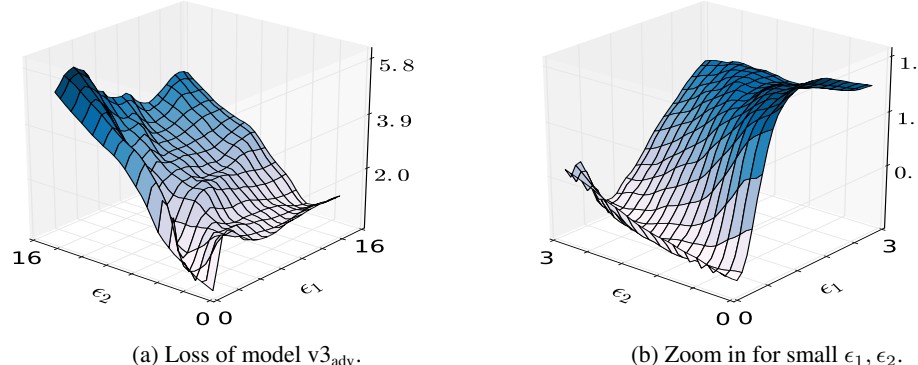

(a) Loss of model v3$_{adv}$.          (b) Zoom in for small $\epsilon_1, \epsilon_2$.

Figure 1: **Gradient masking in single-step adversarial training.** We plot the loss of model v3$_{adv}$ on points $x^* = x + \epsilon_1 \cdot g + \epsilon_2 \cdot g^{\perp}$, where $g$ is the signed gradient and $g^{\perp}$ is an orthogonal adversarial direction. Plot (b) is a zoom of (a) near $x$. The gradient poorly approximates the global loss.

We give a formal statement of this result and of the assumptions on $\mathcal{A}^*$ in Appendix B. Of course, ideally we would like guarantees against *arbitrary* future adversaries. For very low-dimensional tasks (e.g., MNIST), stronger guarantees are within reach for specific classes of adversaries (e.g., $\ell_\infty$ bounded perturbations (Madry et al., 2017; Sinha et al., 2018; Raghunathan et al., 2018; Kolter & Wong, 2017)), yet they also fail to extend to other adversaries not considered at training time (see Appendix C.1 for a discussion). For ImageNet-scale tasks, stronger formal guarantees appear out of reach, and we thus resort to an experimental assessment of the robustness of Ensemble Adversarially Trained models to various non-interactive black-box adversaries in Section 4.2.

## 4 EXPERIMENTS

We show the existence of a degenerate minimum, as described in Section 3.3, for the adversarially trained Inception v3 model of Kurakin et al. (2017b). Their model (denoted v3$_{adv}$) was trained on a Step-LL attack with $\epsilon \leq 16/256$. We also adversarially train an Inception ResNet v2 model (Szegedy et al., 2016a) using the same setup. We denote this model by IRv2$_{adv}$. We refer the reader to (Kurakin et al., 2017b) for details on the adversarial training procedure.

We first measure the *approximation-ratio* of the Step-LL attack for the inner maximization in (1). As we do not know the true maximum, we lower-bound it using an iterative attack. For 1,000 random test points, we find that for a standard Inception v3 model, step-LL gets within 19% of the optimum loss on average. This attack is thus a good candidate for adversarial training. Yet, for the v3$_{adv}$ model, the approximation ratio drops to 7%, confirming that the learned model is less amenable to linearization. We obtain similar results for Inception ResNet v2 models. The ratio is 17% for a standard model, and 8% for IRv2$_{adv}$. Similarly, we look at the *cosine similarity* between the perturbations given by a single-step and multi-step attack. The more linear the model, the more similar we expect both perturbations to be. The average similarity drops from 0.13 for Inception v3 to 0.02 for v3$_{adv}$. This effect is not due to the decision surface of v3$_{adv}$ being "too flat" near the data points: the average gradient norm is larger for v3$_{adv}$ (0.17) than for the standard v3 model (0.10).

We visualize this "gradient-masking" effect (Papernot et al., 2016b) by plotting the loss of v3$_{adv}$ on examples $x^* = x + \epsilon_1 \cdot g + \epsilon_2 \cdot g^{\perp}$, where $g$ is the signed gradient of model v3$_{adv}$ and $g^{\perp}$ is a signed vector orthogonal to $g$. Looking forward to Section 4.1, we actually chose $g^{\perp}$ to be the signed gradient of another Inception model, from which adversarial examples transfer to v3$_{adv}$. Figure 1 shows that the loss is highly curved in the vicinity of the data point $x$, and that the gradient poorly reflects the global loss landscape. Similar plots for additional data points are in Figure 4.

We show similar results for adversarially trained MNIST models in Appendix C.2. On this task, *input dropout* (Srivastava et al., 2014) mitigates adversarial training's overfitting problem, in some cases. Presumably, the random input mask *diversifies* the perturbations seen during training (dropout at intermediate layers does not mitigate the overfitting effect). Mishkin et al. (2017) find that input dropout significantly degrades accuracy on ImageNet, so we did not include it in our experiments.

Table 1: **Error rates (in %) of adversarial examples transferred between models.** We use Step-LL with $\epsilon = {}^{16}/_{256}$ for 10,000 random test inputs. Diagonal elements represent a white-box attack. The best attack for each target appears in bold. Similar results for MNIST models appear in Table 7.

| | | | Source | | | | | | Source | | |
|---|---|---|---|---|---|---|---|---|---|---|---|
| **Target** | v4 | v3 | v3$_{adv}$ | IRv2 | IRv2$_{adv}$ | **Target** | v4 | v3 | v3$_{adv}$ | IRv2 | IRv2$_{adv}$ |
| v4 | **60.2** | 39.2 | 31.1 | 36.6 | 30.9 | v4 | **31.0** | 14.9 | 10.2 | 13.6 | 9.9 |
| v3 | 43.8 | **69.6** | 36.4 | 42.1 | 35.1 | v3 | 18.7 | **42.7** | 13.0 | 17.8 | 12.8 |
| v3$_{adv}$ | **36.3** | 35.6 | ~~26.6~~ | 35.2 | 35.9 | v3$_{adv}$ | 13.6 | 13.5 | ~~9.0~~ | 13.0 | **14.5** |
| IRv2 | 38.0 | 38.0 | 30.8 | **50.7** | 31.9 | IRv2 | 14.1 | 14.8 | 9.9 | **24.0** | 10.6 |
| IRv2$_{adv}$ | **31.0** | 30.3 | 25.7 | 30.6 | ~~21.4~~ | IRv2$_{adv}$ | 10.3 | **10.5** | 7.7 | 10.4 | ~~5.8~~ |
| | | | **Top 1** | | | | | | **Top 5** | | |

## 4.1 ATTACKS AGAINST ADVERSARIALLY TRAINED NETWORKS

Kurakin et al. (2017b) found their adversarially trained model to be robust to various single-step attacks. They conclude that this robustness should translate to attacks *transferred* from other models. As we have shown, the robustness to single-step attacks is actually misleading, as the model has learned to *degrade* the information contained in the model's gradient. As a consequence, we find that the v3$_{adv}$ model is substantially more vulnerable to single-step attacks than Kurakin et al. (2017b) predicted, both in a white-box and black-box setting. The same holds for the IRv2$_{adv}$ model.

In addition to the v3$_{adv}$ and IRv2$_{adv}$ models, we consider standard Inception v3, Inception v4 and Inception ResNet v2 models. These models are available in the `TensorFlow-Slim` library (Abadi et al., 2015). We describe similar results for a variety of models trained on MNIST in Appendix C.2.

**Black-box attacks.** Table 1 shows error rates for single-step attacks transferred between models. We compute perturbations on one model (the source) and transfer them to all others (the targets). When the source and target are the same, the attack is white-box. Adversarial training greatly increases robustness to white-box single-step attacks, but incurs a higher error rate in a black-box setting. Thus, the robustness gain observed when evaluating defended models in isolation is misleading. Given the ubiquity of this pitfall among proposed defenses against adversarial examples (Carlini & Wagner, 2017a; Brendel & Bethge, 2017; Papernot et al., 2016b), we advise researchers *to always consider both white-box and black-box adversaries when evaluating defensive strategies*. Notably, a similar discrepancy between white-box and black-box attacks was recently observed in Buckman et al. (2018).

Attacks crafted on adversarial models are found to be weaker even against undefended models (i.e., when using v3$_{adv}$ or IRv2$_{adv}$ as source, the attack transfers with lower probability). This confirms our intuition from Section 3.3: adversarial training does not just overfit to perturbations that affect standard models, but actively degrades the linear approximation underlying the single-step attack.

**A new randomized single-step attack.** The loss function visualization in Figure 1 shows that sharp curvature artifacts localized near the data points can mask the true direction of steepest ascent. We thus suggest to prepend single-step attacks by a small random step, in order to "escape" the non-smooth vicinity of the data point before linearizing the model's loss. Our new attack, called R+FGSM (alternatively, R+Step-LL), is defined as follows, for parameters $\epsilon$ and $\alpha$ (where $\alpha < \epsilon$):

$$x^{\text{adv}} = x' + (\varepsilon - \alpha) \cdot \texttt{sign}\left(\nabla_{x'} J(x', y_{\text{true}})\right), \quad \text{where} \quad x' = x + \alpha \cdot \texttt{sign}(\mathcal{N}(\mathbf{0}^d, \mathbf{I}^d)) . \quad (7)$$

Note that the attack requires a single gradient computation. The R+FGSM is a computationally efficient alternative to iterative methods that have high success rates in a white-box setting. Our attack can be seen as a single-step variant of the general PGD method from (Madry et al., 2017).

Table 2 compares error rates for the Step-LL and R+Step-LL methods (with $\epsilon = 16/256$ and $\alpha = \epsilon/2$). The extra random step yields a stronger attack for all models, even those without adversarial training. This suggests that a model's loss function is generally less smooth near the data points. We further compared the R+Step-LL attack to a two-step Iter-LL attack, which computes two gradient steps. Surprisingly, we find that for the adversarially trained Inception v3 model, the R+Step-LL attack is *stronger* than the two-step Iter-LL attack. That is, the local gradients learned by the adversarially trained model are *worse than random directions* for finding adversarial examples!

Table 2: **Error rates (in %) for Step-LL, R+Step-LL and a two-step Iter-LL on ImageNet.** We use $\epsilon = {}^{16}\!/{}_{256}$, $\alpha = {}^{\epsilon}\!/{}_{2}$ on 10,000 random test inputs. R+FGSM results on MNIST are in Table 7.

| | v4 | v3 | v3$_{adv}$ | IRv2 | IRv2$_{adv}$ | v4 | v3 | v3$_{adv}$ | IRv2 | IRv2$_{adv}$ |
|---|---|---|---|---|---|---|---|---|---|---|
| **Step-LL** | 60.2 | 69.6 | 26.6 | 50.7 | 21.4 | 31.0 | 42.7 | 9.0 | 24.0 | 5.8 |
| **R+Step-LL** | 70.5 | 80.0 | **64.8** | 56.3 | 37.5 | 42.8 | 57.1 | **37.1** | 29.3 | 15.0 |
| **Iter-LL(2)** | **78.5** | **86.3** | ~~58.3~~ | **69.9** | **41.6** | **56.2** | **70.2** | ~~29.6~~ | **45.4** | **16.5** |
| | | | **Top 1** | | | | | **Top 5** | | |

Table 3: **Models used for Ensemble Adversarial Training on ImageNet.** The ResNets (He et al., 2016) use either 50 or 101 layers. IncRes stands for Inception ResNet (Szegedy et al., 2016a).

| Trained Model | Pre-trained Models | Holdout Models |
|---|---|---|
| Inception v3 (v3$_{adv-ens3}$) | Inception v3, ResNet v2 (50) | Inception v4 |
| Inception v3 (v3$_{adv-ens4}$) | Inception v3, ResNet v2 (50), IncRes v2 | ResNet v1 (50) |
| IncRes v2 (IRv2$_{adv-ens}$) | Inception v3, IncRes v2 | ResNet v2 (101) |

We find that the addition of this random step hinders transferability (see Table 9). We also tried adversarial training using R+FGSM on MNIST, using a similar approach as (Madry et al., 2017). We adversarially train a CNN (model A in Table 5) for 100 epochs, and attain $> 90.0\%$ accuracy on R+FGSM samples. However, training on R+FGSM provides only little robustness to iterative attacks. For the PGD attack of (Madry et al., 2017) with 20 steps, the model attains $18.0\%$ accuracy.

## 4.2 ENSEMBLE ADVERSARIAL TRAINING

We now evaluate our Ensemble Adversarial Training strategy described in Section 3.4. We recall our intuition: by augmenting training data with adversarial examples crafted from *static pre-trained models*, we decouple the generation of adversarial examples from the model being trained, so as to avoid the degenerate minimum described in Section 3.3. Moreover, our hope is that robustness to attacks transferred from some fixed set of models will generalize to other black-box adversaries.

We train Inception v3 and Inception ResNet v2 models (Szegedy et al., 2016a) on ImageNet, using the pre-trained models shown in Table 3. In each training batch, we rotate the source of adversarial examples between the currently trained model and one of the pre-trained models. We select the source model *at random* in each batch, to diversify examples across epochs. The pre-trained models' gradients can be precomputed for the full training set. *The per-batch cost of Ensemble Adversarial Training is thus lower than that of standard adversarial training*: using our method with $n-1$ pre-trained models, only every $n^{\text{th}}$ batch requires a forward-backward pass to compute adversarial gradients. We use synchronous distributed training on 50 machines, with minibatches of size 16 (we did not pre-compute gradients, and thus lower the batch size to fit all models in memory). Half of the examples in a minibatch are replaced by Step-LL examples. As in Kurakin et al. (2017b), we use RMSProp with a learning rate of 0.045, decayed by a factor of 0.94 every two epochs.

To evaluate how robustness to black-box attacks generalizes across models, we transfer various attacks crafted on three different holdout models (see Table 3), as well as on an *ensemble* of these models (as in Liu et al. (2017)). We use the **Step-LL**, **R+Step-LL**, **FGSM**, **I-FGSM** and the **PGD** attack from Madry et al. (2017) using the hinge-loss function from Carlini & Wagner (2017a). Our results are in Table 4. For each model, we report the *worst-case* error rate over all black-box attacks transferred from each of the holdout models (20 attacks in total). Results for MNIST are in Table 8.

**Convergence speed.** Convergence of Ensemble Adversarial Training is slower than for standard adversarial training, a result of training on "hard" adversarial examples and lowering the batch size. Kurakin et al. (2017b) report that after 187 epochs (150$k$ iterations with minibatches of size 32), the v3$_{adv}$ model achieves 78% accuracy. Ensemble Adversarial Training for models v3$_{adv-ens3}$ and v3$_{adv-ens4}$ converges after 280 epochs (450$k$ iterations with minibatches of size 16). The Inception ResNet v2 model is trained for 175 epochs, where a baseline model converges at around 160 epochs.

Table 4: **Error rates (in %) for Ensemble Adversarial Training on ImageNet.** Error rates on clean data are computed over the full test set. For 10,000 random test set inputs, and $\epsilon = {^{16}/_{256}}$, we report error rates on white-box Step-LL and the *worst-case error* over a series of black-box attacks (*Step-LL, R+Step-LL, FGSM, I-FGSM, PGD*) transferred from the holdout models in Table 3. For both architectures, we mark methods tied for best in bold (based on 95% confidence).

| Model | Top 1 | | | Top 5 | | |
|---|---|---|---|---|---|---|
| | **Clean** | **Step-LL** | **Max. Black-Box** | **Clean** | **Step-LL** | **Max. Black-Box** |
| v3 | **22.0** | 69.6 | 51.2 | **6.1** | 42.7 | 24.5 |
| v3$_{adv}$ | **22.0** | **26.6** | 40.8 | **6.1** | **9.0** | 17.4 |
| v3$_{adv-ens3}$ | 23.6 | 30.0 | **34.0** | 7.6 | **10.1** | **11.2** |
| v3$_{adv-ens4}$ | 24.2 | 43.3 | **33.4** | 7.8 | 19.4 | **10.7** |
| IRv2 | **19.6** | 50.7 | 44.4 | **4.8** | 24.0 | 17.8 |
| IRv2$_{adv}$ | **19.8** | **21.4** | 34.5 | **4.9** | **5.8** | 11.7 |
| IRv2$_{adv-ens}$ | **20.2** | 26.0 | **27.0** | **5.1** | 7.6 | **7.9** |

**White-box attacks.** For both architectures, the models trained with Ensemble Adversarial Training are slightly less accurate on clean data, compared to standard adversarial training. Our models are also more vulnerable to white-box single-step attacks, as they were only partially trained on such perturbations. Note that for v3$_{adv-ens4}$, the proportion of white-box Step-LL samples seen during training is $^1/4$ (instead of $^1/3$ for model v3$_{adv-ens3}$). The negative impact on the robustness to white-box attacks is large, for only a minor gain in robustness to transferred samples. Thus it appears that while increasing the *diversity* of adversarial examples seen during training can provide some marginal improvement, the main benefit of Ensemble Adversarial Training is in *decoupling* the attacks from the model being trained, which was the goal we stated in Section 3.4.

Ensemble Adversarial Training is not robust to *white-box* Iter-LL and R+Step-LL samples: the error rates are similar to those for the v3$_{adv}$ model, and omitted for brevity (see Kurakin et al. (2017b) for Iter-LL attacks and Table 2 for R+Step-LL attacks). Kurakin et al. (2017b) conjecture that larger models are needed to attain robustness to such attacks. Yet, against black-box adversaries, these attacks are only a concern insofar as they reliably transfer between models.

**Black-box attacks.** Ensemble Adversarial Training significantly boosts robustness to *all* attacks transferred from the holdout models. For the IRv2$_{adv-ens}$ model, the accuracy loss (compared to IRv2's accuracy on clean data) is **7.4% (top 1)** and **3.1% (top 5)**. We find that the strongest attacks in our test suite (i.e., with highest transfer rates) are the FGSM attacks. Black-box R+Step-LL or iterative attacks are less effective, as they do not transfer with high probability (see Kurakin et al. (2017b) and Table 9). Attacking an ensemble of all three holdout models, as in Liu et al. (2017), did not lead to stronger black-box attacks than when attacking the holdout models individually.

Our results have little variance with respect to the attack parameters (e.g., smaller $\epsilon$) or to the use of other holdout models for black-box attacks (e.g., we obtain similar results by attacking the v3$_{adv-ens3}$ and v3$_{adv-ens4}$ models with the IRv2 model). We also find that v3$_{adv-ens3}$ is not vulnerable to perturbations transferred from v3$_{adv-ens4}$. We obtain similar results on MNIST (see Appendix C.2), thus demonstrating the applicability of our approach to different datasets and model architectures.

**The NIPS 2017 competition on adversarial examples.** Our Inception ResNet v2 model was included as a baseline defense in the NIPS 2017 competition on Adversarial Examples (Kurakin et al., 2017c). Participants of the attack track submitted non-interactive black-box attacks that produce adversarial examples with bounded $\ell_\infty$ norm. Models submitted to the defense track were evaluated on all attacks over a subset of the ImageNet test set. The score of a defense was defined as the *average* accuracy of the model over all adversarial examples produced by all attacks.

Our IRv2$_{adv-ens}$ model *finished 1$^{st}$ among* 70 *submissions* in the first development round, with a score of 95.3% (the second placed defense scored 89.9%). The test data was intentionally chosen as an "easy" subset of ImageNet. Our model achieved 97.9% accuracy on the clean test data.

After the first round, we released our model publicly, which enabled other users to launch white-box attacks against it. Nevertheless, a majority of the final submissions built upon our released model. The winning submission (team "liaofz" with a score of 95.3%) made use of a novel adversarial

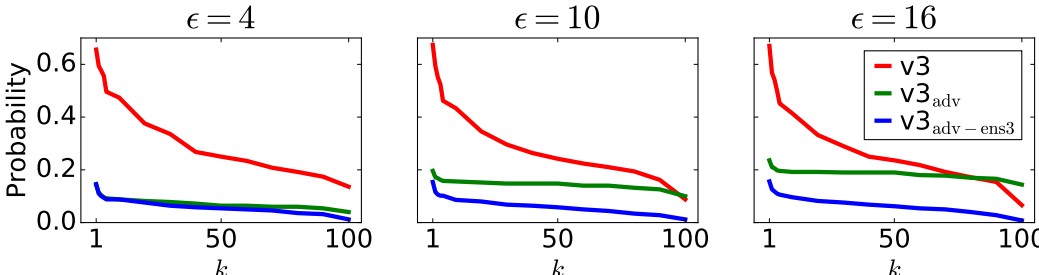

Figure 2: **The dimensionality of the adversarial cone**. For $500$ correctly classified points $x$, and for $\epsilon \in \{4, 10, 16\}$, we plot the probability that we find *at least* $k$ orthogonal vectors $r_i$ such that $\|r_i\|_\infty = \epsilon$ and $x + r_i$ is misclassified. For $\epsilon \geq 10$, model $v3_{adv}$ shows a *bimodal* phenomenon: most points $x$ either have $0$ adversarial directions or more than $90$.

denoising technique. The second placed defense (team "cihangxie" with a score of $92.4\%$) prepends our IRv2$_{adv\text{-}ens}$ model with random padding and resizing of the input image (Xie et al., 2018).

It is noteworthy that the defenses that incorporated Ensemble Adversarial Training faired better against the *worst-case* black-box adversary. Indeed, although very robust on average, the winning defense achieved as low as $11.8\%$ accuracy on some attacks. The best defense under this metric (team "rafaelmm" which randomly perturbed images before feeding them to our IRv2$_{adv\text{-}ens}$ model) achieved at least $53.6\%$ accuracy *against all submitted attacks*, including the attacks that explicitly targeted our released model in a white-box setting.

**Decreasing gradient masking.**  Ensemble Adversarial Training decreases the magnitude of the gradient masking effect described previously. For the $v3_{adv\text{-}ens3}$ and $v3_{adv\text{-}ens4}$ models, we find that the loss incurred on a Step-LL attack gets within respectively $13\%$ and $18\%$ of the optimum loss (we recall that for models $v3$ and $v3_{adv}$, the approximation ratio was respectively $19\%$ and $7\%$). Similarly, for the IRv2$_{adv\text{-}ens}$ model, the ratio improves from $8\%$ (for IRv2$_{adv}$) to $14\%$. As expected, not solely training on a white-box single-step attack reduces gradient masking. We also verify that after Ensemble Adversarial Training, a two-step iterative attack outperforms the R+Step-LL attack from Section 4.1, thus providing further evidence that these models have meaningful gradients.

Finally, we revisit the "Gradient-Aligned Adversarial Subspace" (GAAS) method of Tramèr et al. (2017). Their method estimates the size of the space of adversarial examples in the vicinity of a point, by finding a set of *orthogonal* perturbations of norm $\epsilon$ that are all adversarial. We note that adversarial perturbations do not technically form a "subspace" (e.g., the $\mathbf{0}$ vector is not adversarial). Rather, they may form a "cone", the dimension of which varies as we increase $\epsilon$. By linearizing the loss function, estimating the dimensionality of this cone reduces to finding vectors $r_i$ that are strongly aligned with the model's gradient $g = \nabla_x L(h(x), y_{\text{true}})$. Tramèr et al. (2017) give a method that finds $k$ orthogonal vectors $r_i$ that satisfy $g^\top r_i \geq \epsilon \cdot \|g\|_2 \cdot \frac{1}{\sqrt{k}}$ (this bound is tight). We extend this result to the $\ell_\infty$ norm, an open question in Tramèr et al. (2017). In Section E, we give a randomized combinatorial construction (Colbourn, 2010), that finds $k$ orthogonal vectors $r_i$ satisfying $\|r_i\|_\infty = \epsilon$ and $\mathbb{E}\left[g^\top r_i\right] \geq \epsilon \cdot \|g\|_1 \cdot \frac{1}{\sqrt{k}}$. We show that this result is tight as well.

For models $v3$, $v3_{adv}$ and $v3_{adv\text{-}ens3}$, we select $500$ correctly classified test points. For each $x$, we search for a maximal number of orthogonal adversarial perturbations $r_i$ with $\|r_i\|_\infty = \epsilon$. We limit our search to $k \leq 100$ directions per point. The results are in Figure 2. For $\epsilon \in \{4, 10, 16\}$, we plot the proportion of points that have at least $k$ orthogonal adversarial perturbations. For a fixed $\epsilon$, the value of $k$ can be interpreted as the dimension of a "slice" of the cone of adversarial examples near a data point. For the standard Inception v3 model, we find over $50$ orthogonal adversarial directions for $30\%$ of the points. The $v3_{adv}$ model shows a curious bimodal phenomenon for $\epsilon \geq 10$: for most points ($\approx 80\%$), we find no adversarial direction aligned with the gradient, which is consistent with the gradient masking effect. Yet, for most of the remaining points, the adversarial space is very high-dimensional ($k \geq 90$). Ensemble Adversarial Training yields a more robust model, with only a small fraction of points near a large adversarial space.

## 5 CONCLUSION AND FUTURE WORK

Previous work on adversarial training at scale has produced encouraging results, showing strong robustness to (single-step) adversarial examples (Goodfellow et al., 2014b; Kurakin et al., 2017b). Yet, these results are misleading, as the adversarially trained models remain vulnerable to simple black-box and white-box attacks. Our results, generic with respect to the application domain, suggest that adversarial training can be improved by *decoupling* the generation of adversarial examples from the model being trained. Our experiments with Ensemble Adversarial Training show that the robustness attained to attacks from some models *transfers* to attacks from other models.

We did not consider black-box adversaries that attack a model via other means than by transferring examples from a local model. For instance, generative techniques (Baluja & Fischer, 2017) might provide an avenue for stronger attacks. Yet, a recent work by Xiao et al. (2018) found Ensemble Adversarial Training to be resilient to such attacks on MNIST and CIFAR10, and often attaining higher robustness than models that were adversarially trained on iterative attacks.

Moreover, *interactive* adversaries (see Appendix A) could try to exploit queries to the target model's prediction function in their attack, as demonstrated in Papernot et al. (2017). If queries to the target model yield *prediction confidences*, an adversary can estimate the target's gradient at a given point (e.g., using finite-differences as in Chen et al. (2017)) and fool the target with our R+FGSM attack. Note that if queries only return the predicted label, the attack does not apply. Exploring the impact of these classes of black-box attacks and evaluating their scalability to complex tasks is an interesting avenue for future work.

## ACKNOWLEDGMENTS

We thank Ben Poole and Jacob Steinhardt for feedback on early versions of this work. Nicolas Papernot is supported by a Google PhD Fellowship in Security. Research was supported in part by the Army Research Laboratory, under Cooperative Agreement Number W911NF-13-2-0045 (ARL Cyber Security CRA), and the Army Research Office under grant W911NF-13-1-0421. The views and conclusions contained in this document are those of the authors and should not be interpreted as representing the official policies, either expressed or implied, of the Army Research Laboratory or the U.S. Government. The U.S. Government is authorized to reproduce and distribute reprints for government purposes notwithstanding any copyright notation hereon.

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

## A   THREAT MODEL: FORMAL DEFINITIONS

We provide formal definitions for the threat model introduced in Section 3.1. In the following, we explicitly identify the hypothesis space $\mathcal{H}$ that a model belongs to as describing the model's *architecture*. We consider a target model $h \in \mathcal{H}$ trained over inputs $(x, y_{\text{true}})$ sampled from a data distribution $\mathcal{D}$. More precisely, we write

$$h \leftarrow \texttt{train}(\mathcal{H}, X_{\text{train}}, Y_{\text{train}}, r) \,,$$

where $\texttt{train}$ is a *randomized training procedure* that takes in a description of the model architecture $\mathcal{H}$, a training set $X_{\text{train}}, Y_{\text{train}}$ sampled from $\mathcal{D}$, and randomness $r$.

Given a set of test inputs $X, Y = \{(x_1, y_1), \ldots, (x_m, y_m)\}$ from $\mathcal{D}$ and a budget $\epsilon > 0$, an adversary $\mathcal{A}$ produces adversarial examples $X^{\text{adv}} = \{x_1^{\text{adv}}, \ldots, x_m^{\text{adv}}\}$, such that $\|x_i - x_i^{\text{adv}}\|_\infty \leq \epsilon$ for all $i \in [1, m]$. We evaluate success of the attack as the error rate of the target model over $X^{\text{adv}}$:

$$\frac{1}{m} \sum_{i=1}^{m} \mathbb{1}(\arg\max h(x_i^{\text{adv}}) \neq y_i) \,.$$

We assume $\mathcal{A}$ can sample inputs according to the data distribution $\mathcal{D}$. We define three adversaries.

**Definition 2** (White-Box Adversary). *For a target model $h \in \mathcal{H}$, a white-box adversary is given access to all elements of the training procedure, that is $\texttt{train}$ (the training algorithm), $\mathcal{H}$ (the model architecture), the training data $X_{train}, Y_{train}$, the randomness $r$ and the parameters $h$. The adversary can use any attack (e.g., those in Section 3.2) to find adversarial inputs.*

White-box access to the internal model weights corresponds to a very strong adversarial model. We thus also consider the following relaxed and arguably more realistic notion of a *black-box* adversary.

**Definition 3** (*Non-Interactive* Black-Box Adversary). *For a target model $h \in \mathcal{H}$, a non-interactive black-box adversary only gets access to $\texttt{train}$ (the target model's training procedure) and $\mathcal{H}$ (the model architecture). The adversary can sample from the data distribution $\mathcal{D}$, and uses a local algorithm to craft adversarial examples $X^{adv}$.*

Attacks based on transferability (Szegedy et al., 2013) fall in this category, wherein the adversary selects a procedure $\texttt{train}'$ and model architecture $\mathcal{H}'$, trains a local model $h'$ over $\mathcal{D}$, and computes adversarial examples on its local model $h'$ using white-box attack strategies.

Most importantly, a black-box adversary does not learn the randomness $r$ used to train the target, nor the target's parameters $h$. The black-box adversaries in our paper are actually slightly *stronger* than the ones defined above, in that they use the same training data $X_{\text{train}}, Y_{\text{train}}$ as the target model.

We provide $\mathcal{A}$ with the target's training procedure $\texttt{train}$ to capture knowledge of *defensive strategies* applied at training time, e.g., adversarial training (Szegedy et al., 2013; Goodfellow et al., 2014b) or ensemble adversarial training (see Section 4.2). For ensemble adversarial training, $\mathcal{A}$ also knows the architectures of all pre-trained models. In this work, we always mount black-box attacks that train a local model with a *different* architecture than the target model. We actually find that black-box attacks on adversarially trained models are stronger in this case (see Table 1).

The main focus of our paper is on non-interactive black-box adversaries as defined above. For completeness, we also formalize a stronger notion of *interactive* black-box adversaries that additionally issue prediction queries to the target model (Papernot et al., 2017). We note that in cases where ML models are deployed as part of a larger system (e.g., a self driving car), an adversary may not have direct access to the model's query interface.

**Definition 4** (*Interactive* Black-Box Adversary). *For a target model $h \in \mathcal{H}$, an interactive black-box adversary only gets access to $\texttt{train}$ (the target model's training procedure) and $\mathcal{H}$ (the model architecture). The adversary issues (adaptive) oracle queries to the target model. That is, for arbitrary inputs $x \in [0, 1]^d$, the adversary obtains $y = \arg\max h(x)$ and uses a local algorithm to craft adversarial examples (given knowledge of $\mathcal{H}$, $\texttt{train}$, and tuples $(x, y)$).*

Papernot et al. (2017) show that such attacks are possible even if the adversary only gets access to a small number of samples from $\mathcal{D}$. Note that if the target model's prediction interface additionally returns class scores $h(x)$, interactive black-box adversaries could use queries to the target model to estimate the model's gradient (e.g., using finite differences) (Chen et al., 2017), and then apply the attacks in Section 3.2. We further discuss interactive black-box attack strategies in Section 5.

## B  Generalization Bound for ensemble Adversarial Training

We provide a formal statement of Theorem 1 in Section 3.4, regarding the generalization guarantees of Ensemble Adversarial Training. For simplicity, we assume that the model is trained solely on adversarial examples computed on the pre-trained models (i.e., we ignore the clean training data and the adversarial examples computed on the model being trained). Our results are easily extended to also consider these data points.

Let $\mathcal{D}$ be the data distribution and $\mathcal{A}_1, \ldots, \mathcal{A}_k, \mathcal{A}^*$ be adversarial distributions where a sample $(x, y)$ is obtained by sampling $(x, y_{\text{true}})$ from $\mathcal{D}$, computing an $x^{\text{adv}}$ such that $\|x^{\text{adv}} - x\|_\infty \leq \epsilon$ and returning $(x^{\text{adv}}, y_{\text{true}})$. We assume the model is trained on $N$ data points $Z_{\text{train}}$, where $\frac{N}{k}$ data points are sampled from each distribution $\mathcal{A}_i$, for $1 \leq i \leq k$. We denote $\mathcal{A}_{\text{train}} = \{\mathcal{A}_1, \ldots, \mathcal{A}_k\}$. At test time, the model is evaluated on adversarial examples from $\mathcal{A}^*$.

For a model $h \in \mathcal{H}$ we define the empirical risk

$$\hat{R}(h, \mathcal{A}_{\text{train}}) := \frac{1}{N} \sum_{(x^{\text{adv}}, y_{\text{true}}) \in Z_{\text{train}}} L(h(x^{\text{adv}}), y_{\text{true}}) , \tag{8}$$

and the risk over the target distribution (or future adversary)

$$R(h, \mathcal{A}^*) := \mathop{\mathbb{E}}_{(x^{\text{adv}}, y_{\text{true}}) \sim \mathcal{A}^*} [L(h(x^{\text{adv}}), y_{\text{true}})] . \tag{9}$$

We further define the average *discrepancy distance* (Mansour et al., 2009) between distributions $\mathcal{A}_i$ and $\mathcal{A}^*$ with respect to a hypothesis space $\mathcal{H}$ as

$$\text{disc}_{\mathcal{H}}(\mathcal{A}_{\text{train}}, \mathcal{A}^*) := \frac{1}{k} \sum_{i=1}^{k} \sup_{h_1, h_2 \in \mathcal{H}} \left| \mathop{\mathbb{E}}_{\mathcal{A}_i} \left[ \mathbb{1}_{\{h_1(x^{\text{adv}}) = h_2(x^{\text{adv}})\}} \right] - \mathop{\mathbb{E}}_{\mathcal{A}^*} \left[ \mathbb{1}_{\{h_1(x^{\text{adv}}) = h_2(x^{\text{adv}})\}} \right] \right| . \tag{10}$$

This quantity characterizes how "different" the future adversary is from the train-time adversaries. Intuitively, the distance $\text{disc}(\mathcal{A}_{\text{train}}, \mathcal{A}^*)$ is small if the difference in robustness between two models to the target attack $\mathcal{A}^*$ is somewhat similar to the difference in robustness between these two models to the attacks used for training (e.g., if the static black-box attacks $\mathcal{A}_i$ induce much higher error on some model $h_1$ than on another model $h_2$, then the same should hold for the target attack $\mathcal{A}^*$). In other words, the *ranking* of the robustness of models $h \in \mathcal{H}$ should be similar for the attacks in $\mathcal{A}_{\text{train}}$ as for $\mathcal{A}^*$.

Finally, let $R_N(\mathcal{H})$ be the average *Rademacher complexity* of the distributions $\mathcal{A}_1, \ldots, \mathcal{A}_k$ (Zhang et al., 2012). Note that $R_N(\mathcal{H}) \to 0$ as $N \to \infty$. The following theorem is a corollary of Zhang et al. (2012, Theorem 5.2):

**Theorem 5.** *Assume that $\mathcal{H}$ is a function class consisting of bounded functions. Then, with probability at least $1 - \epsilon$,*

$$\sup_{h \in \mathcal{H}} |\hat{R}(h, \mathcal{A}_{train}) - R(h, \mathcal{A}^*)| \leq disc_{\mathcal{H}}(\mathcal{A}_{train}, \mathcal{A}^*) + 2R_N(\mathcal{H}) + O\left(\sqrt{\frac{\ln(1/\epsilon)}{N}}\right) . \tag{11}$$

Compared to the standard generalization bound for supervised learning, the generalization bound for Domain Adaptation incorporates the extra term $\text{disc}_{\mathcal{H}}(\mathcal{A}_{\text{train}}, \mathcal{A}^*)$ to capture the divergence between the target and source distributions. In our context, this means that the model $h^*$ learned by Ensemble Adversarial Training has guaranteed generalization bounds with respect to future adversaries that are not "too different" from the ones used during training. Note that $\mathcal{A}^*$ need not restrict itself to perturbation with bounded $\ell_\infty$ norm for this result to hold.

## C  Experiments on MNIST

We re-iterate our ImageNet experiments on MNIST. For this simpler task, Madry et al. (2017) show that training on iterative attacks conveys robustness to white-box attacks with bounded $\ell_\infty$ norm. Our goal is not to attain similarly strong white-box robustness on MNIST, but to show that our observations on limitations of single-step adversarial training, extend to other datasets than ImageNet.

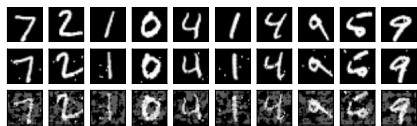

Figure 3: **Adversarial Examples on MNIST.** (top) clean examples. (middle) inputs are rotated by 20° and 5 random pixels are flipped. (bottom) The I-FGSM with $\epsilon = 0.3$ is applied.

## C.1   A NOTE ON $\ell_\infty$ ROBUSTNESS ON MNIST

The MNIST dataset is a simple baseline for assessing the potential of a defense, but the obtained results do not always generalize to harder tasks. We suggest that this is because achieving robustness to $\ell_\infty$ perturbations admits a simple "closed-form" solution, given the near-binary nature of the data. Indeed, for an average MNIST image, over $80\%$ of the pixels are in $\{0, 1\}$ and only $6\%$ are in the range $[0.2, 0.8]$. Thus, for a perturbation with $\epsilon \le 0.3$, binarized versions of $x$ and $x^{\mathrm{adv}}$ can differ in at most $6\%$ of the input dimensions. By binarizing the inputs of a standard CNN trained *without adversarial training*, we obtain a model that enjoys robustness similar to the model trained by Madry et al. (2017). Concretely, for a white-box I-FGSM attack, we get at most $11.4\%$ error.

The existence of such a simple robust representation begs the question of why learning a robust model with adversarial training takes so much effort. Finding techniques to improve the performance of adversarial training, even on simple tasks, could provide useful insights for more complex tasks such as ImageNet, where we do not know of a similarly simple "denoising" procedure.

These positive results on MNIST for the $\ell_\infty$ norm also leave open the question of defining a general norm for adversarial examples. Let us motivate the need for such a definition: we find that if we first *rotate* an MNIST digit by 20°, and then use the I-FGSM, our rounding model and the model from Madry et al. (2017) achieve only $65\%$ accuracy (on "clean" rotated inputs, the error is $< 5\%$). If we further randomly "flip" 5 pixels per image, the accuracy of both models drops to under $50\%$. Thus, we successfully evade the model by slightly extending the threat model (see Figure 3).

Of course, we could augment the training set with such perturbations (see Engstrom et al. (2017)). *An open question is whether we can enumerate all types of "adversarial" perturbations.* In this work, we focus on the $\ell_\infty$ norm to illustrate our findings on the limitations of single-step adversarial training on ImageNet and MNIST, and to showcase the benefits of our Ensemble Adversarial Training variant. Our approach can easily be extended to consider multiple perturbation metrics. We leave such an evaluation to future work.

## C.2   RESULTS

We repeat experiments from Section 4 on MNIST. We use the architectures in Table 5. We train a standard model for 6 epochs, and an adversarial model with the FGSM ($\epsilon = 0.3$) for 12 epochs.

During adversarial training, we avoid the *label leaking* effect described by Kurakin et al. (2017b) by using the model's predicted class $\arg \max h(x)$ instead of the true label $y_{\mathrm{true}}$ in the FGSM,

We first analyze the "degenerate" minimum of adversarial training, described in Section 3.3. For each trained model, we compute the *approximation-ratio* of the FGSM for the inner maximization problem in equation (1). That is, we compare the loss produced by the FGSM with the loss of a

Table 5: **Neural network architectures used in this work for the MNIST dataset.** Conv: convolutional layer, FC: fully connected layer.

| A | B | C | D |
|---|---|---|---|
| Conv(64, 5, 5) + Relu | Dropout(0.2) | Conv(128, 3, 3) + Tanh | $\begin{bmatrix} \text{FC(300) + Relu} \\ \text{Dropout(0.5)} \end{bmatrix} \times 4$ |
| Conv(64, 5, 5) + Relu | Conv(64, 8, 8) + Relu | MaxPool(2,2) | FC + Softmax |
| Dropout(0.25) | Conv(128, 6, 6) + Relu | Conv(64, 3, 3) + Tanh | |
| FC(128) + Relu | Conv(128, 5, 5) + Relu | MaxPool(2,2) | |
| Dropout(0.5) | Dropout(0.5) | FC(128) + Relu | |
| FC + Softmax | FC + Softmax | FC + Softmax | |

Table 6: **Approximation ratio between optimal loss and loss induced by single-step attack on MNIST.** Architecture B' is the same as B without the input dropout layer.

| A | $A_{adv}$ | B | $B_{adv}$ | $B^*$ | $B^*_{adv}$ | C | $C_{adv}$ | D | $D_{adv}$ |
|---|---|---|---|---|---|---|---|---|---|
| 17% | 0% | 25% | 8% | 23% | 1% | 25% | 0% | 49% | 16% |

strong iterative attack. The results appear in Table 6. As we can see, for all model architectures, adversarial training degraded the quality of a linear approximation to the model's loss.

We find that *input dropout* (Srivastava et al., 2014) (i.e., randomly dropping a fraction of input features during training) as used in architecture B limits this unwarranted effect of adversarial training.[2] If we omit the input dropout (we call this architecture $B^*$) the single-step attack degrades significantly. We discuss this effect in more detail below. For the fully connected architecture D, we find that the learned model is very close to linear and thus also less prone to the degenerate solution to the min-max problem, as we postulated in Section 3.3.

**Attacks.** Table 7 compares error rates of undefended and adversarially trained models on white-box and black-box attacks, as in Section 4.1. Again, model B presents an anomaly. For all other models, we corroborate our findings on ImageNet for adversarial training: (1) black-box attacks trump white-box single-step attacks; (2) white-box single-step attacks are significantly stronger if prepended by a random step. For model $B_{adv}$, the opposite holds true. We believe this is because input dropout increases diversity of attack samples similarly to Ensemble Adversarial Training.

Table 7: **White-box and black-box attacks against standard and adversarially trained models.** For each model, the strongest single-step white-box and black box attacks are marked in bold.

| | white-box | | black-box | | | | |
|---|---|---|---|---|---|---|---|
| | FGSM | R+FGSM | $FGSM_A$ | $FGSM_B$ | $FGSM_{B^*}$ | $FGSM_C$ | $FGSM_D$ |
| A | 64.7 | **69.7** | - | **61.5** | 53.2 | 46.8 | 41.5 |
| $A_{adv}$ | 2.2 | **14.8** | 6.6 | **10.7** | 8.8 | 6.5 | 8.3 |
| B | 85.0 | **86.0** | 45.7 | - | 69.9 | 59.9 | **85.9** |
| $B_{adv}$ | **11.6** | 11.1 | 6.4 | **8.9** | 8.5 | 4.9 | 6.1 |
| $B^*$ | **75.7** | 74.1 | 44.3 | **72.8** | - | 46.0 | 62.6 |
| $B^*_{adv}$ | 4.3 | **40.6** | 16.1 | 14.7 | 15.0 | **17.9** | 9.1 |
| C | **81.8** | **81.8** | 40.2 | 55.8 | 49.5 | - | **59.4** |
| $C_{adv}$ | 3.7 | **17.1** | 9.8 | **29.3** | 21.5 | 11.9 | 21.9 |
| D | 92.4 | **95.4** | 61.3 | **74.1** | 68.9 | 65.1 | - |
| $D_{adv}$ | 25.5 | **47.5** | **32.1** | 30.5 | 29.3 | 28.2 | 21.8 |

While training with input dropout helps avoid the degradation of the single-step attack, it also significantly delays convergence of the model. Indeed, model $B_{adv}$ retains relatively high error on white-box FGSM examples. Adversarial training with input dropout can be seen as comparable to training with a randomized single-step attack, as discussed in Section 4.1.

The positive effect of input dropout is architecture and dataset specific: Adding an input dropout layer to models A, C and D confers only marginal benefit, and is outperformed by Ensemble Adversarial Training, discussed below. Moreover, Mishkin et al. (2017) find that input dropout significantly degrades accuracy on ImageNet. We thus did not incorporate it into our models on ImageNet.

**Ensemble Adversarial Training.** To evaluate Ensemble Adversarial Training 3.4, we train two models per architecture. The first, denoted $[A-D]_{adv-ens}$, uses a single pre-trained model of the same type (i.e., $A_{adv-ens}$ is trained on perturbations from another model A). The second model, denoted $[A-D]_{adv-ens3}$, uses 3 pre-trained models ($\{A, C, D\}$ or $\{B, C, D\}$). We train all models for 12 epochs.

We evaluate our models on black-box attacks crafted on models A,B,C,D (for a fair comparison, we do not use the same pre-trained models for evaluation, but retrain them with different random seeds).

---

[2]We thank Arjun Bhagoji, Bo Li and Dawn Song for this observation.

Table 8: **Ensemble Adversarial Training on MNIST.** For black-box robustness, we report the maximum and average error rate over a suite of 12 attacks, comprised of the FGSM, I-FGSM and PGD (Madry et al., 2017) attacks applied to models A,B,C and D. We use $\epsilon = 16$ in all cases. For each model architecture, we mark the models tied for best (at a $95\%$ confidence level) in bold.

|  | Clean | FGSM | Max. Black Box | Avg. Black Box |
|---|---|---|---|---|
| $A_{adv}$ | **0.8** | **2.2** | 10.8 | 7.7 |
| $A_{adv\text{-}ens}$ | **0.8** | 7.0 | **6.6** | 5.2 |
| $A_{adv\text{-}ens3}$ | **0.7** | 5.4 | **6.5** | **4.3** |
| $B_{adv}$ | **0.8** | **11.6** | 8.9 | **5.5** |
| $B_{adv\text{-}ens}$ | **0.7** | **10.5** | **6.8** | **5.3** |
| $B_{adv\text{-}ens3}$ | **0.8** | 14.0 | 8.8 | **5.1** |
| $C_{adv}$ | **1.0** | 3.7 | 29.3 | 18.7 |
| $C_{adv\text{-}ens}$ | **1.3** | **1.9** | 17.2 | 10.7 |
| $C_{adv\text{-}ens3}$ | **1.4** | 3.6 | **14.5** | **8.4** |
| $D_{adv}$ | **2.6** | 25.5 | 32.5 | 23.5 |
| $D_{adv\text{-}ens}$ | **2.6** | **21.5** | 38.6 | 28.0 |
| $D_{adv\text{-}ens3}$ | **2.6** | 29.4 | **29.8** | **15.6** |

The attacks we consider are the FGSM, I-FGSM and the PGD attack from Madry et al. (2017) with the loss function from Carlini & Wagner (2017a)), all with $\epsilon = 0.3$. The results appear in Table 8. For each model, we report the worst-case and average-case error rate over all black-box attacks.

Ensemble Adversarial Training significantly increases robustness to black-box attacks, except for architecture B, which we previously found to not suffer from the same overfitting phenomenon that affects the other adversarially trained networks. Nevertheless, model $B_{adv\text{-}ens}$ achieves slightly better robustness to white-box and black-box attacks than $B_{adv}$. In the majority of cases, we find that using a single pre-trained model produces good results, but that the extra diversity of including three pre-trained models can sometimes increase robustness even further. Our experiments confirm our conjecture that robustness to black-box attacks generalizes across models. Indeed, we find that when training with three external models, we attain very good robustness against attacks initiated from models with the same architecture (as evidenced by the average error on our attack suite), but also increased robustness to attacks initiated from the fourth holdout model

## D  TRANSFERABILITY OF RANDOMIZED SINGLE-STEP PERTURBATIONS.

In Section 4.1, we introduced the R+Step-LL attack, an extension of the Step-LL method that prepends the attack with a small random perturbation. In Table 9, we evaluate the transferability of R+Step-LL adversarial examples on ImageNet. We find that the randomized variant produces perturbations that transfer at a much lower rate (see Table 1 for the deterministic variant).

Table 9: **Error rates (in $\%$) of randomized single-step attacks transferred between models on ImageNet.** We use R+Step-LL with $\epsilon = 16/256, \alpha = \epsilon/2$ for 10,000 random test set samples. The white-box attack always outperforms black-box attacks.

| | | | Source | | | | | | Source | | |
|---|---|---|---|---|---|---|---|---|---|---|---|
| Target | v4 | v3 | $v3_{adv}$ | IRv2 | $IRv2_{adv}$ | Target | v4 | v3 | $v3_{adv}$ | IRv2 | $IRv2_{adv}$ |
| v4 | **70.5** | 37.2 | 23.2 | 34.0 | 24.6 | v4 | **42.8** | 14.3 | 6.3 | 11.9 | 6.9 |
| v3 | 42.6 | **80.0** | 26.7 | 38.5 | 27.6 | v3 | 18.0 | **57.1** | 8.0 | 15.6 | 8.6 |
| $v3_{adv}$ | 31.4 | 30.7 | **64.8** | 30.4 | 34.0 | $v3_{adv}$ | 10.7 | 10.4 | **37.1** | 10.1 | 12.9 |
| IRv2 | 36.2 | 35.7 | 23.0 | **56.3** | 24.6 | IRv2 | 12.8 | 13.6 | 6.1 | **29.3** | 7.0 |
| $IRv2_{adv}$ | 26.8 | 26.3 | 25.2 | 26.9 | **37.5** | $IRv2_{adv}$ | 8.0 | 8.0 | 7.7 | 8.3 | **15.0** |
| | | | Top 1 | | | | | | Top 5 | | |

# E    GRADIENT ALIGNED ADVERSARIAL SUBSPACES FOR THE $\ell_\infty$ NORM

Tramèr et al. (2017) consider the following task for a given model $h$: for a (correctly classified) point $x$, find $k$ orthogonal vectors $\{r_1, \ldots, r_k\}$ such that $\|r_i\|_2 \leq \epsilon$ and all the $x + r_i$ are adversarial (i.e., $\arg\max h(x + r_i) \neq y_{\text{true}}$). By linearizing the model's loss function, this reduces to finding $k$ orthogonal vectors $r_i$ that are maximally aligned with the model's gradient $g = \nabla_x L(h(x), y_{\text{true}})$. Tramèr et al. (2017) left a construction for the $\ell_\infty$ norm as an open problem.

We provide an optimal construction for the $\ell_\infty$ norm, based on *Regular Hadamard Matrices* (Colbourn, 2010). Given the $\ell_\infty$ constraint, we find orthogonal vectors $r_i$ that are maximally aligned with the *signed* gradient, $\text{sign}(g)$. We first prove an analog of (Tramèr et al., 2017, Lemma 1).

**Lemma 6.** *Let $v \in \{-1, 1\}^d$ and $\alpha \in (0, 1)$. Suppose there are $k$ orthogonal vectors $r_1, \ldots r_n \in \{-1, 1\}^d$ satisfying $v^\top r_i \geq \alpha \cdot d$. Then $\alpha \leq k^{-\frac{1}{2}}$.*

*Proof.* Let $\hat{r}_i = \frac{r_i}{\|r_i\|_2} = \frac{r_i}{\sqrt{d}}$. Then, we have

$$d = \|v\|_2^2 \geq \sum_{i=1}^k |v^\top \hat{r}_i|^2 = d^{-1} \sum_{i=1}^k |v^\top r_i|^2 \geq d^{-1} \cdot k \cdot (\alpha \cdot d)^2 = k \cdot \alpha^2 \cdot d \,, \qquad (12)$$

from which we obtain $\alpha \leq k^{-\frac{1}{2}}$. $\qquad\square$

This result bounds the number of orthogonal perturbations we can expect to find, for a given alignment with the signed gradient. As a warm-up consider the following trivial construction of $k$ orthogonal vectors in $\{-1, 1\}^d$ that are "somewhat" aligned with $\text{sign}(g)$. We split $\text{sign}(g)$ into $k$ "chunks" of size $\frac{d}{k}$ and define $r_i$ to be the vector that is equal to $\text{sign}(g)$ in the $i^{\text{th}}$ chunk and zero otherwise. We obtain $\text{sign}(g)^\top r_i = \frac{d}{k}$, a factor $\sqrt{k}$ worse than the the bound in Lemma 6.

We now provide a construction that meets this upper bound. We make use of Regular Hadamard Matrices of order $k$ (Colbourn, 2010). These are square matrices $H_k$ such that: (1) all entries of $H_k$ are in $\{-1, 1\}^k$; (2) the rows of $H_k$ are mutually orthogonal; (3) All row sums are equal to $\sqrt{k}$.

The order of a Regular Hadamard Matrix is of the form $4u^2$ for an integer $u$. We use known constructions for $k \in \{4, 16, 36, 64, 100\}$.

**Lemma 7.** *Let $g \in \mathbb{R}^d$ and $k$ be an integer for which a Regular Hadamard Matrix of order $k$ exists. Then, there is a randomized construction of $k$ orthogonal vectors $r_1, \ldots r_n \in \{-1, 1\}^d$, such that $\text{sign}(g)^\top r_i = d \cdot k^{-1/2}$. Moreover, $\mathbb{E}[g^\top r_i] = k^{-1/2} \cdot \|g\|_1$.*

*Proof.* We construct $k$ orthogonal vectors $r_1, \ldots, r_k \in \{-1, 1\}^d$, where $r_i$ is obtained by repeating the i$^{\text{th}}$ row of $H_k$ $d/k$ times (for simplicity, we assume that $k$ divides $d$. Otherwise we pad $r_i$ with zeros). We then multiply each $r_i$ component-wise with $\text{sign}(g)$. By construction, the $k$ vectors $r_i \in \{-1, 1\}^d$ are mutually orthogonal, and we have $\text{sign}(g)^\top r_i = \frac{d}{k} \cdot \sqrt{k} = d \cdot k^{-1/2}$, which is tight according to Lemma 6.

As the weight of the gradient $g$ may not be uniformly distributed among its $d$ components, we apply our construction to a random permutation of the signed gradient. We then obtain

$$\mathbb{E}[g^\top r_i] = \mathbb{E}\Big[ \sum_{j=1}^d |g^{(j)}| \cdot \text{sign}(g^{(j)}) \cdot r_i^{(j)} \Big] \qquad (13)$$

$$= \sum_{j=1}^d |g^{(j)}| \cdot \mathbb{E}\Big[ \text{sign}(g^{(j)}) \cdot r_i^{(j)} \Big] = k^{-1/2} \cdot \|g\|_1 \,. \qquad (14)$$

$\qquad\square$

It can be shown that the bound in Lemma 7 can be attained if and only if the $r_i$ are constructed from the rows of a Regular Hadamard Matrix (Colbourn, 2010). For general integers $k$ for which no such matrix exists, other combinatorial designs may be useful for achieving looser bounds.

# F   ILLUSTRATIONS OF GRADIENT MASKING IN ADVERSARIAL TRAINING

In Section 3.3, we show that adversarial training introduces spurious curvature artifacts in the model's loss function around data points. As a result, one-shot attack strategies based on first-order approximations of the model loss produce perturbations that are non-adversarial. In Figures 4 and 5 we show further illustrations of this phenomenon for the Inception $v3_{adv}$ model trained on ImageNet by Kurakin et al. (2017b) as well as for the model $A_{adv}$ we trained on MNIST.

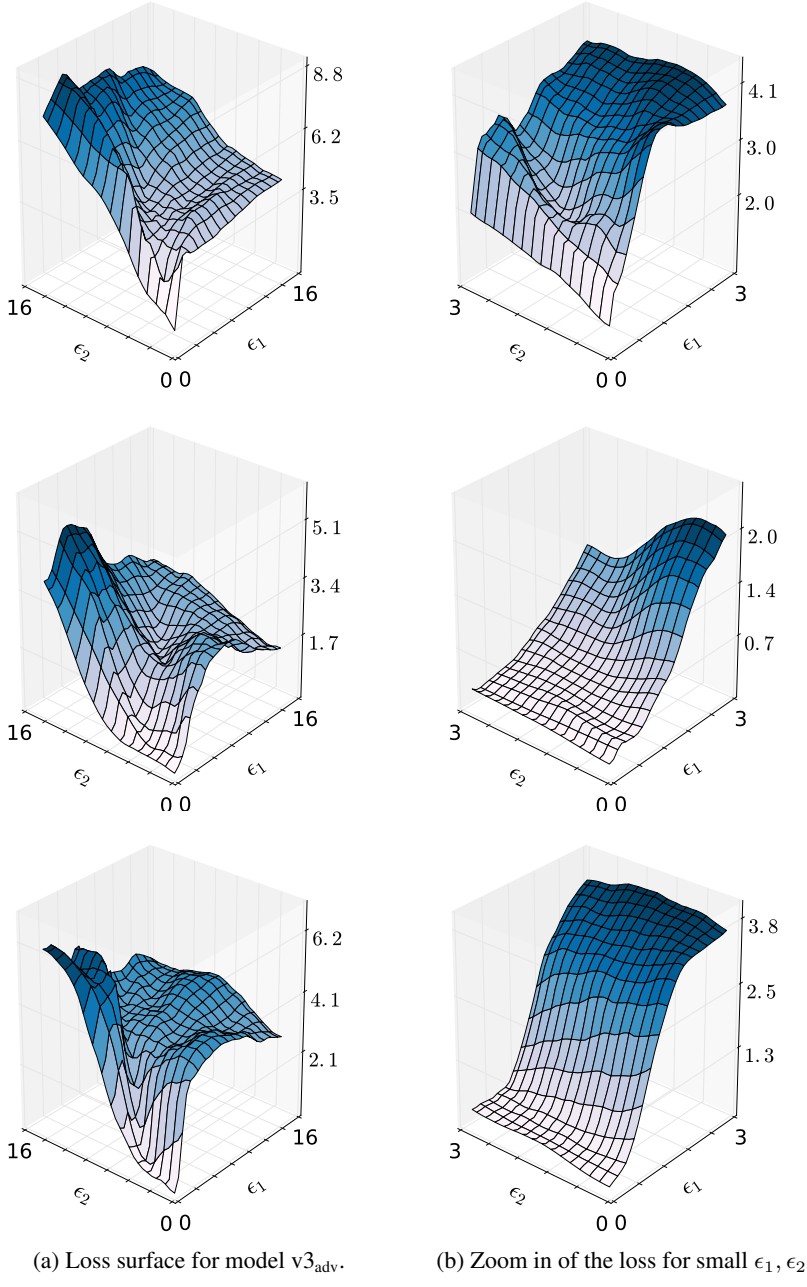

(a) Loss surface for model $v3_{adv}$.          (b) Zoom in of the loss for small $\epsilon_1, \epsilon_2$.

Figure 4: **Additional illustrations of the local curvature artifacts introduced by adversarial training on ImageNet.** We plot the loss of model $v3_{adv}$ on samples of the form $x^* = x + \epsilon_1 \cdot g + \epsilon_2 \cdot g^\perp$, where $g$ is the signed gradient of $v3_{adv}$ and $g^\perp$ is an orthogonal adversarial direction, obtained from an Inception v4 model. The right-side plots are zoomed in versions of the left-side plots.

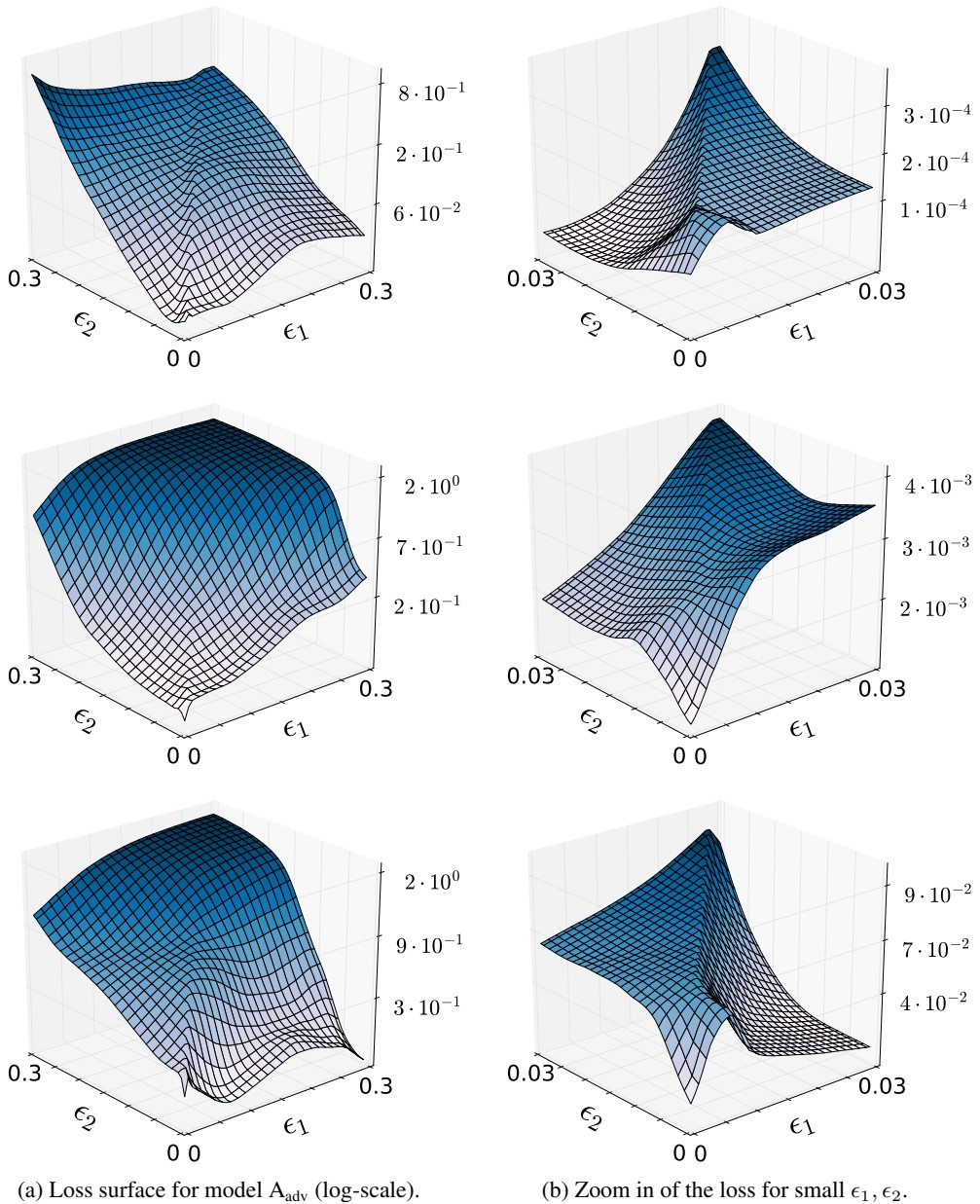

(a) Loss surface for model $A_{adv}$ (log-scale).       (b) Zoom in of the loss for small $\epsilon_1, \epsilon_2$.

Figure 5: **Illustrations of the local curvature artifacts introduced by adversarial training on MNIST.** We plot the loss of model $A_{adv}$ on samples of the form $x^* = x + \epsilon_1 \cdot g + \epsilon_2 \cdot g^\perp$, where $g$ is the signed gradient of model $A_{adv}$ and $g^\perp$ is an orthogonal adversarial direction, obtained from model B. The right-side plots are zoomed in versions of the left-side plots.

