# OpenReview forum: "Ensemble Adversarial Training: Attacks and Defenses"
_ICLR.cc/2018/Conference — Accept (Poster)_

### Official Review · AnonReviewer1 · 2017-11-25
**The ideas are not surprising but seem reasonable and practically useful.**

**Rating:** 6
**Confidence:** 2

**Review:**

This paper proposes ensemble adversarial training, in which adversarial examples crafted on other static pre-trained models are used in the training phase. Their method makes deep networks robust to black-box attacks, which was empirically demonstrated.

This is an empirical paper. The ideas are simple and not surprising but seem reasonable and practically useful.
Empirical results look natural.

[Strong points]
* Proposed randomized white-box attacks are empirically shown to be stronger than original ones.
* Proposed ensemble adversarial training empirically achieves smaller error rate for black-box attacks.

[Weak points]
* no theoretical guarantee for proposed methods.
* Robustness of their ensemble adversarial training depends on what pre-trained models and attacks are used in the training phase.

---

> ### Author Response · Authors · 2017-12-21
> **Thanks for the feedback**
>
> Thank you for the constructive review.
>
> > Robustness of their ensemble adversarial training depends on what pre-trained models and attacks are used in the training phase
>
> While we agree that the robustness of ensemble adversarial training may depend on the choices of model architectures and attacks used, this is not fundamentally different from the meta-parameter choices faced with "regular" adversarial training, or even non-adversarial training. For instance, it has been shown that the choice of model architecture has a strong influence on how well regular adversarial training performs (e.g., see our MNIST experiments in Appendix C.2 of the revised manuscript). For the Inception v3 architecture, we find that ensemble adversarial training with two different sets of pre-trained models yield very similar results.
>
> Regarding the diversity of models used, we note that the main goal of ensemble adversarial training is to decouple the attack from the model being trained, in order to prevent gradient masking. Our MNIST experiments (Appendix C.2 of the revised manuscript) show that using a single pre-trained model with the same architecture than the model being trained is often a very effective form of ensemble adversarial training. We have emphasized the importance of decoupling gradients in our paper.
>
> > no theoretical guarantee for proposed methods.
>
> We thank you for raising the question of formal guarantees for future attacks (indeed our models remain vulnerable to white-box l-infinity attacks). Following your suggestion, we draw a connection between Ensemble Adversarial Training and the formal generalization guarantees obtained for Domain Adaptation, wherein a model is trained on multiple source distributions and evaluated on a different target distribution (Section 3.4 and Appendix B in our revised manuscript). While the resulting bounds may not necessarily be meaningful in practice, they do show that Ensemble Adversarial Training can provide formal guarantees for future adversaries of “similar power” than the ones considered during training. Some works manage to provide stronger guarantees than ours for small datasets (e.g., against all bounded l-infinity attacks), using techniques that appear out of reach for ImageNet-scale tasks. Yet, even extending these guarantees to arbitrary adversaries is a daunting task, given that we do not know how to define or enumerate the right sets of adversarial metrics. We believe that this connection to Domain Adaptation will be interesting to the community, as the resulting bounds are independent of the noise model (e.g., l-infinity perturbations) being considered.

---

### Official Review · AnonReviewer2 · 2017-11-28
**Interesting empirical analysis and heuristics**

**Rating:** 6
**Confidence:** 4

**Review:**

This paper describes computationally efficient methods for training adversarially robust deep neural networks for image classification. (These methods may extend to other machine learning models and domains as well, but that's beyond the scope of this paper.)

The former standard method for generating adversarially images quickly and using them in training was to do a single gradient step to increase the loss of the true label or decrease the loss of an alternate label. This paper shows that such training methods only lead to robustness against these "weak" adversarial examples, leaving the adversarially-trained models vulnerable to multi-step white-box attacks and black-box attacks (adversarial examples generated to attack alternate models).

There are two proposed solutions. The first is to generate additional adversarial examples from other models and use them in training. This seems to yield robustness against black-box attacks from held-out models as well.  Of course, it requires that you have a somewhat diverse group of models to choose from. If that's the case, why not directly build an ensemble of all the models? An ensemble of neural networks can still be represented as a neural network, although a more computationally costly one. Thus, while this heuristic appears to be useful with current models against current attacks, I don't know how well it will hold up in the future.

The second solution is to add random noise before taking the gradient step.  This yields more effective adversarial examples, both for attacking models and for training, because it relies less on the local gradient. This is another simple idea that appears to be effective. However, I would be interested to see a comparison to a 2-step gradient-based attack.  R+Step-LL can be viewed as a 2-step attack: a random step followed by a gradient step. What if both steps were gradient steps instead? This interpolates between Step-LL and I-Step-LL, with an intermediate computational cost. It would be very interesting to know if R+Step-LL is more or less effective than 2+Step-LL, and how large the difference is.

I like that this paper demonstrates the weakness of previous methods, including extensive experiments and a very nice visualization of the loss landscape in two adversarial dimensions. The proposed heuristics seem effective in practice, but they're somewhat ad hoc and there is no analysis of how these heuristics might or might not be vulnerable to future attacks.

---

> ### Author Response · Authors · 2017-12-21
> **Thanks for the feedback**
>
> Thank you for the constructive review.
>
> > Of course, it requires that you have a somewhat diverse group of models to choose from. If that's the case, why not directly build an ensemble of all the models
>
> A large diversity of pre-trained models is not necessary for ensemble adversarial training. The main goal of our approach is to decouple the attack (the method used to produce adversarial examples) from the defense (the model being trained) so as to avoid the gradient masking issue. In this sense, even using a single pre-trained model is valuable and we indeed found this to be very effective on MNIST (Appendix C.2 in our revised manuscript).
> Of course, using multiple models will only increase the diversity of adversarial examples encountered during training. As shown by Liu et al. (ICLR’17), applying the FGSM to different ImageNet models generates very diverse perturbations (the gradients of different models are often close to orthogonal) but these perturbations still transfer between the models. Thus, although different models can produce very diverse attacks, simply ensembling these models is not necessarily a good defense strategy, as the same adversarial examples will fool most of the models in the ensemble. For instance, if we ensemble all the pre-trained ImageNet models we used, except for Inception v4, and then use a black-box FGSM attack computed on Inception v4, the ensemble's robustness is only marginally better than that of a single undefended model.
> When using Ensemble Adversarial Training with the Inception v3 architecture, we found that the marginal benefit of adding more pre-trained models is relatively low, thus also corroborating the fact that the main benefit of Ensemble Adversarial Training is in decoupling the attack procedure from the model being trained. We have clarified this point in our paper.
>
> > It would be very interesting to know if R+Step-LL is more or less effective than 2+Step-LL, and how large the difference is.
>
> We thank the reviewer for the question about the 2-step Iter-LL attack, as it yields another nice illustration of the gradient masking effect. It turns out that for non-defended models and ensemble-adversarially trained models, a 2-step iterative attack is stronger than R+Step-LL, as is to be expected (the difference is roughly 10% top1/top5 accuracy). However, for standard adversarial training on Inception v3, R+Step-LL is stronger than the 2-step Iter-LL attack (by about 7% top1/top5 accuracy). Thus, this shows that the local gradient of the adversarially trained model is worse than a random direction from an optimization perspective. We added these results to Table 2 in our paper.
>
> > The proposed heuristics seem effective in practice, but they're somewhat ad hoc and there is no analysis of how these heuristics might or might not be vulnerable to future attacks
>
> We thank you for raising the question of formal guarantees for future attacks (indeed our models remain vulnerable to white-box l-infinity attacks). Following your suggestion, we draw a connection between Ensemble Adversarial Training and the formal generalization guarantees obtained for Domain Adaptation, wherein a model is trained on multiple source distributions and evaluated on a different target distribution (Section 3.4 and Appendix B in our revised manuscript). While the resulting bounds may not necessarily be meaningful in practice, they do show that Ensemble Adversarial Training can provide formal guarantees for future adversaries of “similar power” than the ones considered during training. Some works manage to provide stronger guarantees than ours for small datasets (e.g., against all bounded l-infinity attacks), using techniques that appear out of reach for ImageNet-scale tasks. Yet, even extending these guarantees to arbitrary adversaries is a daunting task, given that we do not know how to define or enumerate the right sets of adversarial metrics. We believe that this connection to Domain Adaptation will be interesting to the community, as the resulting bounds are independent of the noise model (e.g., l-infinity perturbations) being considered.
>
> There is also an independent submission (https://openreview.net/forum?id=HknbyQbC-) that proposes a different type of black-box attacks based on GANs, that we did not consider in our paper. The authors evaluate their attack against ensemble adversarially trained models and find that our defense outperforms both standard adversarial training, as well as approaches with strong guarantees against white-box robustness, on both MNIST and CIFAR10. This provides further evidence that our defense generalizes to attacks unseen during training (and also that it works well on CIFAR10).

---

### Official Review · AnonReviewer3 · 2017-12-03
**a simple an effective method against static black-box attacks, but it remains unclear whether the models are more robust in general**

**Rating:** 6
**Confidence:** 4

**Review:**

The paper proposes a modification to adversarial training. Instead of alternating between clean and examples generated on-the-fly by the fast gradient sign during training, the model training is performed by alternating clean examples and adversarial examples generated from pre-trained models. The motivation behind this change is that one-step method to generate adversarial examples fail at generating good adversarial examples when applied to models trained in the adversarial setting. In contrast, one-step methods applied to models trained only on natural data generate adversarial examples that transfer reasonably well, even on models trained with usual adversarial training. The authors also propose a slight modification to the fast gradient sign method, in which an adversarial example is created using a random perturbation and the current model's gradient, which seems to work better than the fast gradient sign method. Experiments with inception models on ImageNet show increased robustness both against "black-box" attacks using held-out models not used in ensemble adversarial training.

One advantage of the method is that it is extremely simple. It uses pre-trained models that are readily available, and gains robustness against several well-known adversaries widely considered in the state of the art. The experiments are carried out on ImageNet and are seriously conducted.

On the negative side, there is a significant loss in accuracy, and the models are more vulnerable to white-box attacks than using standard adversarial training. As the authors discuss in the conclusion, this leaves open the question as to whether the models are indeed more robust, or whether it is an artifact of the static black-box attack schemes that are considered in the paper, which measures how much a single model is robust to adversarial examples for other models that were trained independently. For instance, there are no experiments against what is called adaptive black-box adversaries; one could also imagine finding adversarial examples that are trained to fool all models in a predefined collection of models. In the end, while the work presented in the paper found its use in the recent NIPS competition on defending against adversarial examples, it is still unclear whether this kind of defence would make a difference in critical applications.

---

> ### Author Response · Authors · 2017-12-21
> **Thanks for the feedback**
>
> Thank you for the constructive review.
>
> > On the negative side, there is a significant loss in accuracy
>
> While the drop in accuracy (on ImageNet) is not zero, it is small for the Inception ResNet v2 model (0.6% top1 and 0.3% top5) and somewhat larger for Inception v3 (1.6-2.2% top1/top5). However, there are few other defenses proposed on ImageNet to compare these numbers against. One concurrent submission (https://openreview.net/forum?id=SyJ7ClWCb) also aims at increasing white-box robustness at the cost of a much larger decrease in clean accuracy (10-15% top1).
>
> > the models are more vulnerable to white-box attacks than using standard adversarial training
>
> We would like to clarify that our models are not more vulnerable to white-box attacks than those learned with standard adversarial training. While this appears to be the case on single-step attacks, it is due to the absence of a gradient masking effect, which is an intended consequence of ensemble adversarial training.
> For iterative white-box attacks on ImageNet, we find that the robustness increase with adversarial training (whether the standard version or our ensemble variant) is only marginal compared to standard training. This was already observed in the "Adversarial Training at Scale" paper of Kurakin et al., ICLR'17.
> While there have been some recent successes in hardening models against white-box attacks, the techniques required are expensive and not currently applicable to large-scale problems such as ImageNet. Incidentally, an independent submission (https://openreview.net/forum?id=HyydRMZC-) shows that ensemble adversarial training is more robust than other adversarial training variants against white-box “spatially transformed” adversarial examples.
>
> > one could also imagine finding adversarial examples that are trained to fool all models in a predefined collection of models
>
> Thank you for bringing this to our attention, we have updated our manuscript to clarify that we evaluated our models against adversarial examples that evade a collection of models. Specifically, we applied various attacks (including multi-step attacks like Step-LL, Iter-LL, PGD, etc.) to an ensemble of all of our holdout models on ImageNet (Inception V4, ResNet v1 and ResNet v2) and then transferred these examples to the adversarially trained models. We did not find this to produce a stronger attack and have clarified this point in our paper.
>
> > there are no experiments against what is called adaptive black-box adversaries
>
> For adaptive attacks, there are few baselines to evaluate defenses against. The “substitute model” attack of Papernot et al. (https://arxiv.org/abs/1602.02697) is hard to scale to ImageNet (this was attempted in https://arxiv.org/abs/1708.03999). For MNIST, Papernot et al. report that their attack is mostly ineffective against an adversarially trained model.
> There is also a concurrent submission that proposes an adaptive attack that attempts to “crawl” the model’s decision boundary (https://openreview.net/forum?id=SyZI0GWCZ). The attack requires a large number of calls to the black-box model (~10,000 per image) and is optimized for the l2 metric. We have attempted to transpose this attack to the l-infinity metric considered in our paper, but have not yet been able to find a set of hyperparameters that produces adversarial examples with small perturbations (even for undefended models). Further work in this direction will be very valuable for the community, and we believe our ensemble adversarially trained models can serve as a good baseline for evaluating new attacks.
>
> For instance, ensemble adversarial training is used a baseline in an independent submission (https://openreview.net/forum?id=HknbyQbC-) which considers black-box attacks based on GANs. The authors find that our defense outperforms both standard adversarial training, as well as approaches with strong guarantees against white-box robustness, on both MNIST and CIFAR10. This provides further evidence that our defense generalizes to unseen attacks (and also that it works well on CIFAR10).
>
> Finally, from a formal perspective, we discovered a natural connection between Ensemble Adversarial Training and Domain Adaptation, wherein a model is trained on multiple source distributions and evaluated on a different target distribution. Generalization bounds obtained in that litterature transfer to our setting, and allow us to express some formal guarantees for future adversaries that are not significantly more powerful than the ones considered during training (Section 3.4 and Appendix B in our revised manuscript).
> Although these bounds are not as strong as some of the formal guarantees obtained for simpler tasks, we believe these results and this connection will be interesting to the community, as they are independent of the noise model (e.g., l-infinity perturbations) being considered.

---

### Author Response · Authors · 2017-12-21
**Impact and follow up work**

Our work, which was available on arxiv this year, has already inspired follow up work from independent authors. In particular, the ensemble adversarially trained models that we publically released during the NIPS competition on adversarial defenses (https://www.kaggle.com/c/nips-2017-defense-against-adversarial-attack) are already being used in multiple papers as baselines for evaluating attacks and building defenses. We will provide a link to our released models in the final version of our paper.

In addition to the models that we released, we believe that our observations on gradient masking in adversarial training are also very useful to the community. Indeed, prior techniques that exhibited this phenomenon (e.g., distillation) were somewhat more “obvious”, in that the defense technique explicitly promotes flat gradients. Our advice to systematically evaluate defenses on both white-box and black-box attacks (in Section 4.1) is being followed in many recent papers (e.g., https://openreview.net/forum?id=SyJ7ClWCb, https://openreview.net/forum?id=rJzIBfZAb, https://openreview.net/forum?id=S18Su--CW).

Below, we describe some of the papers that build upon our publically-released ensemble adversarially trained models. Note that these papers contain references to an earlier (non-anonymized) arXiv version of our work.

An independent submission to ICLR (https://openreview.net/forum?id=HknbyQbC-, avg. score of 5.2) considers black-box attacks based on GANs. The authors evaluate their attack against ensemble adversarial training and find that our defense outperforms both standard adversarial training, as well as approaches with strong guarantees against white-box robustness, on both MNIST and CIFAR10. This provides further evidence that our defense generalizes to attacks unseen during training (and also that it works well on CIFAR10).

Another independent submission based on our work is https://openreview.net/forum?id=Sk9yuql0Z (avg. score of 6.4). This paper describes the defense that ranked 2nd in the final round of the recent NIPS competition on adversarial examples. It prepends our publically released ensemble adversarially trained model with randomized input transformations (image resizing and padding). The authors show that these transformations boost the robustness of the base model they are applied to, and are thus particularly effective when combined with ensemble adversarial training.

Finally, a majority of the top-placed teams in the NIPS competition used similar strategies: they extended our ensemble adversarially trained models using techniques such as ensembling, randomized transforms, image compression, etc. We have added more information on the competition results in Section 4.2. The principal take-away is that defenses that built upon Ensemble Adversarial Training attained high robustness even against the strongest black-box attacks submitted to the competition.

---

### Decision · Program_Chairs · 2018-01-29
**ICLR 2018 Conference Acceptance Decision**

**Decision:**

Accept (Poster)

**Comment:**

The paper studies a defense against adversarial examples that re-trains convolutional networks on adversarial examples constructed to attack pre-trained networks. Whilst the proposed approach is not very original, the paper does present a solid empirical baseline for these kinds of defenses. In particular, it goes beyond the "toy" experiments that most other studies in this space perform by experimenting on ImageNet. This is important as there is evidence suggesting that defenses against adversarial examples that work well on MNIST/CIFAR do not necessarily transfer well to ImageNet. The importance of the baseline method studied in this paper is underlined by its frequent application in the recent NIPS competition on adversarial examples.

---

> ### Public Comment · (anonymous) · 2018-02-01
> **Seeking a reference**
>
> "This is important as there is evidence suggesting that defenses against adversarial examples that work well on MNIST/CIFAR do not necessarily transfer well to ImageNet" is there a reference supporting this? I'm interested in this topic, and would be curious to know!